# Rebound Inverts the *Staphylococcus aureus* Bacteremia Prevention Effect of Antibiotic Based Decontamination Interventions in ICU Cohorts with Prolonged Length of Stay

**DOI:** 10.3390/antibiotics13040316

**Published:** 2024-03-29

**Authors:** James Hurley

**Affiliations:** 1Melbourne Medical School, University of Melbourne, Melbourne, VIC 3052, Australia; jamesh@bhs.org.au; 2Ballarat Health Services, Grampians Health, Ballarat, VIC 3350, Australia; 3Ballarat Clinical School, Deakin University, Ballarat, VIC 3350, Australia

**Keywords:** *Staphylococcus aureus*, bacteremia, antibiotic-based decontamination, selective digestive decontamination, spill-over, rebound, meta-regression, structural equation model, intensive care unit

## Abstract

Could rebound explain the paradoxical lack of prevention effect against *Staphylococcus aureus* blood stream infections (BSIs) with antibiotic-based decontamination intervention (BDI) methods among studies of ICU patients within the literature? Two meta-regression models were applied, each versus the group mean length of stay (LOS). Firstly, the prevention effects against *S. aureus* BSI [and *S. aureus* VAP] among 136 studies of antibiotic-BDI versus other interventions were analyzed. Secondly, the *S. aureus* BSI [and *S. aureus* VAP] incidence in 268 control and intervention cohorts from studies of antibiotic-BDI versus that among 165 observational cohorts as a benchmark was modelled. In model one, the meta-regression line versus group mean LOS crossed the null, with the antibiotic-BDI prevention effect against *S. aureus* BSI at mean LOS day 7 (OR 0.45; 0.30 to 0.68) inverted at mean LOS day 20 (OR 1.7; 1.1 to 2.6). In model two, the meta-regression line versus group mean LOS crossed the benchmark line, and the predicted *S. aureus* BSI incidence for antibiotic-BDI groups was 0.47; 0.09–0.84 percentage points below versus 3.0; 0.12–5.9 above the benchmark in studies with 7 versus 20 days mean LOS, respectively. Rebound within the intervention groups attenuated and inverted the prevention effect of antibiotic-BDI against *S. aureus* VAP and BSI, respectively. This explains the paradoxical findings.

## 1. Introduction

Patient colonization [1,2], ICU colonization pressure [3,4] and length of stay (LOS) are each major risk factors for ICU-acquired infection with *S. aureus*. In European ICU cohorts, *S. aureus* colonization increases the risk of *S. aureus* pneumonia by up to 15-fold, with pneumonia onset generally within 14 days following ICU admission [1,2].

Reducing both patient and ICU colonization, through decontamination interventions, would seem logical to prevent *S. aureus* infections, whether occurring as pneumonia or blood stream infections (BSIs), and multiple potential candidate agents have been tested in various singleton and combination regimens [5,6]. However, the results of numerous studies of decontamination interventions conducted in ICU populations are unclear at four levels [7,8].

Firstly, at the patient level, the strong prevention effects seen against overall ventilator-associated pneumonia (VAP) and overall BSI in antibiotic-based decontamination intervention (BDI) studies, and possibly also antiseptic-BDI studies, generally match the prevention effects against *S. aureus* VAP but, paradoxically, not against *S. aureus* BSI [9,10,11,12,13,14].

Second, at the level of the ICU population, concerns for decontamination possibly ‘driving’ the emergence of both antimicrobial resistance and *S. aureus* and other Gram-positive infections with its prolonged use in the ICU, and rebound colonization and infection on withdrawal of antibiotic-based decontamination interventions (BDIs), remain unresolved [15,16,17,18,19,20,21,22].

Thirdly, there is the unresolved issue of spillover effects to concurrent patients within the ICU not receiving decontamination which might also ‘drive’ event rates [17,18,19]. This is especially concerning for high quality studies of decontamination interventions with concurrent controls [23].

Finally, decontamination interventions presumably mediate their prevention effects against *S. aureus* infections by targeted decolonization of *S. aureus*. Hence, modelling of *S. aureus* colonization would provide more accurate insights into the various ‘drivers’ within these studies, particularly where the prophylactic measures do not eradicate colonizing staphylococci completely.

There are four objectives here. Firstly, to recapitulate indicative prevention effect size estimates for antibiotic-BDI versus other interventions against both overall VAP and overall BSI, as well as *S. aureus* VAP and *S. aureus* BSI end points within the literature [9,10,11,12,13,24,25,26,27,28,29,30,31,32,33,34,35,36,37]. Secondly, to determine whether the study level prevention effect sizes vary with study LOS. The third objective is to detect rebound in incidence of *S. aureus* VAP and *S. aureus* BSIs within these studies versus that within ICU patient cohorts and within study arms not exposed, either directly or indirectly, to decontamination interventions. To this end, meta-regression and structural equation modelling of *S. aureus* colonization are each applied to the data from studies in the broader literature which serve, ‘in toto’, as a natural experiment of various exposures. The fourth objective is to triangulate the findings here with the previous effect size summaries for antiseptic- and antibiotic-BDIs within the literature.

## 2. Results

### 2.1. Characteristics of the Studies

Of the 282 studies identified by the search, 61 of 139 intervention studies were found in either ten Cochrane or other systematic reviews [9,10,11,12,13,24,25,26,27,28,29,30,31,32,33,34,35,36,37]. Most studies were published between 1990 and 2010 and most had a mean ICU-LOS exceeding ten days. Fourteen studies of infection prevention interventions had more than one type of intervention group and fourteen studies had either more than one or no control group. Most groups from observational studies had more than 150 patients per group versus less than 150 patients in the groups of the interventional studies. Most studies originated from either North American or European ICUs. Both *S. aureus* VAP and *S. aureus* BSI incidence data were available from 32 studies, while only *S. aureus* BSI incidence data were available from 34 studies and only *S. aureus* VAP incidence data from the remaining 215 studies (Table 1).

### 2.2. Indicative Effect Sizes

The indicative study-specific and summary effect sizes for the three categories of intervention against *S. aureus* VAP and *S. aureus* BSI are presented as caterpillar plots (Appendix A). Significant summary prevention effects against both overall VAP and against *S. aureus* VAP were evident for all three categories. A summary prevention effect against overall BSI was apparent for the antiseptic-BDI and antibiotic-BDI categories but no category demonstrated a summary prevention effect against *S. aureus* BSI (Table 1). Of note, of the 25 studies of antibiotic-BDI against *S. aureus* BSI, the study specific odds ratios were <0.5 for seven and >2.0 for six (Appendix A).

### 2.3. Prevention Effect Size Meta-Regression versus LOS

The relationship between *S. aureus* VAP prevention effect sizes versus group mean LOS for the three categories of interventions are presented as meta-regressions (see Appendix A; Appendix A). In these models, all three categories of intervention demonstrated significant prevention effects against *S. aureus* VAP at the day 7 intercept but in each case the effect attenuated towards the null in association with increasing group mean LOS.

The relationship between *S. aureus* BSI effect sizes versus group mean LOS for the three categories of interventions are presented as meta-regressions (see Appendix A and Appendix A). In these models, only the antibiotic-BDI category demonstrated significant prevention effects against *S. aureus* BSI at the day 7 intercept. For all three categories, there was attenuation of prevention effects against *S. aureus* BSI [manifested as a positive slope coefficient] in association with increasing group mean LOS which, in the case of the antiseptic-BDI (Figure 1a) and antibiotic-BDI (Figure 1b) categories, crossed the null. Consequently, the predicted antibiotic-BDI prevention effect against *S. aureus* BSI at day 7 (OR 0.45; 0.30 to 0.68) inverts at day 20 (OR 1.7; 1.1 to 2.6).

### 2.4. S. aureus VAP and BSI Incidence versus LOS

The incidence of *S. aureus* VAP and *S. aureus* BSI across all categories varied in each case by >100-fold. The *S. aureus* VAP and *S. aureus* BSI data versus group mean LOS as derived from the meta-regression models for each category in comparison to the benchmark groups are displayed (Table 2 and Appendix A, and Figure 2, Appendix A).

The predicted *S. aureus* VAP for benchmark groups per 100 patients with a group mean LOS of 7 and 20 days was 3.0 (2.5–3.6) and 7.6 (5.9–9.1), respectively (Figure 2 and Appendix A). The predicted *S. aureus* BSI for benchmark groups with a group mean LOS of 7 and 20 days was 1.1 (0.9–1.5) and 3.2 (1.9–4.7), respectively (Figure 2 and Appendix A and Table 2).

The day 7 predicted *S. aureus* VAP per 100 patients for the antiseptic-BDI and antibiotic-BDI intervention groups were each approximately 1.5 percentage points below that predicted for benchmark groups, whereas the day 20 predicted *S. aureus* VAP for the antibiotic-BDI and antiseptic-BDI intervention groups were each 1.5 to 2.5 percentage points (*p* = NS) above that predicted for benchmark groups. By contrast, the predicted *S. aureus* VAP for antibiotic-BDI control groups was approximately 4 percentage points above that predicted for benchmark groups at both day 7 and day 20 (Table 2).

The day 7 predicted *S. aureus* BSI per 100 patients for the antibiotic-BDI intervention groups was 0.4 percentage points below that predicted for the benchmark groups whereas the day 20 predicted *S. aureus* BSI for the antibiotic-BDI intervention groups was >2 percentage points above that predicted for not only the benchmark groups (being double that of the benchmark groups) but also that predicted for all other control and intervention groups (Table 2).

### 2.5. GSEM Modelling of S. aureus Colonization

The GSEM model (Figure 3 and Table 3) is based on the postulated model of causation wherein *S. aureus* colonization, a latent variable in the model, ‘drives’ the count of *S. aureus* VAP and *S. aureus* BSI. *S. aureus* colonization in turn is ‘driven’ by increasing LOS, membership of a concurrent control group (in response to spillover), and origin from a trauma ICU as positive factors and exposure to antibiotic-BDI and antiseptic-BDI as negative factors. Adding the interaction terms between increasing LOS and exposure to antiseptic-and antibiotic-BDI, which represent rebound, lowers the Akaike Information Criteria (AIC), which indicates improvement to the model.

## 3. Discussion

In this analysis of the results from 282 studies of three broad categories of infection prevention interventions, there were four objectives. Firstly, in contrast to the non-decontamination interventions, both antiseptic-BDI and antibiotic-BDI have strong prevention activities against both overall and *S. aureus* VAP, and indeed overall BSI, but neither have activity in preventing *S. aureus* BSI in summary estimates. The indicative summary prevention effect size estimates here recapitulate previous estimates in the literature based on fewer studies (Table 4).

Secondly, attenuation of prevention activity is generally apparent across all categories of intervention and for both *S. aureus* infection end points in association with increasing mean study LOS. However, antibiotic-BDI prevention activity against *S. aureus* BSI attenuates, crosses the null and inverts in studies with LOS > 14 days. This inversion could account for the otherwise paradoxical finding of apparent lack of effect of antibiotic-BDI against *S. aureus* BSI in the summary effect size estimates derived across all studies as generally reported (Table 4).

Thirdly, the *S. aureus* BSI incidence rebounds among the antibiotic-BDI groups with LOS >14 days to exceed that predicted for not only benchmark but also all other categories of the control and intervention groups (Figure 2). This rebound underlies the inversion of the antibiotic-BDI prevention effect against *S. aureus* BSI in studies with mean LOS > 14 days. There are too few studies of antiseptic-BDI to be able to determine whether or when rebound might occur and whether prevention effect inversion occurs.

The postulated prevention effect of antibiotic-BDI (appearing in the model as TAP) against *S. aureus* colonization is affirmed within the GSEM, together with the confounding effects of group mean LOS and rebound (appearing as an interaction term between LOS and antibiotic-BDI appearing as TAP). Rebound is a positive ‘driver’ in this model.

Finally, the findings here can account for rebound and spillover resulting from antibiotic- and antiseptic-BDI and yet the summary effect size estimates derived here are consistent with previous published effect size summaries (Table 4). This triangulation reconciles an outstanding paradox in the literature.

That antibiotic-BDI’s should have strong prevention effects against both overall VAP, overall BSI and *S. aureus* VAP in summary estimates, and yet not against *S. aureus* BSI, is paradoxical (Table 4; Appendix A). This paradox is also reflected in the wide range of study-specific effect sizes of three large individual studies included in this analysis, in which decreased *S. aureus* BSI was [38] or was not observed [39,40] where similar antibiotic-BDIs regimens were used. Moreover, in one large study (>2000 patients per arm) of an antiseptic-BDI regimen [39], the *S. aureus* BSI incidence in the chlorhexidine arm (1.2%) was double that of the control arm (0.6%). Of further note, a narrow spectrum anti-*S. aureus* monoclonal antibody failed to prevent *S. aureus* BSI or *S. aureus* VAP in a large RCT with a mean LOS ~20 days, contrary to the results of preclinical studies [41].

Moreover, antibiotic-BDI intervention groups have higher day 20 incidences of *S. aureus* BSI than control groups (Table 2) among studies which, overall, appear to indicate prevention of *S. aureus* VAP (Table 4). This observation is profoundly paradoxical and represents the effect of rebound which is more apparent for *S. aureus* BSI than for *S. aureus* VAP (Appendix A).

Rebound infection on premature withdrawal of antibiotic-BDI had been noted among patients neutropenic following cytotoxic chemotherapy in hematology units in the 1970s. These severe and occasionally fatal infections were observed in patients who had prematurely discontinued antibiotic-BDI due to its intolerable taste. Rebound sepsis has been noted following hospital discharge among patients exposed to antibiotic therapy considered high risk for causing microbiome disruption [42].

Rebound may be imperceptible without specific surveillance for colonization and infections on withdrawal of decontamination interventions [20,21,22]. Rebound following antibiotic-BDI discontinuation and ICU discharge manifests as a 50% (non-significant) increased risk of hospital-acquired infection [22]. Rebound of ceftazidime-resistant Gram-negative bacteria may occur as a ‘whole of ICU’ phenomenon not limited to the antibiotic-BDI recipients, persisting as an ecological effect for several months after antibiotic-BDI withdrawal [20].

The use of PPAP within some concurrent control groups may have modified the rebound and spillover effects from the intervention groups within some of the antibiotic-BDI studies.

Increased rates of *S. aureus* and other Gram-positive isolates were noted in the five years following the introduction of antibiotic-BDI into Dutch and Austrian ICU’s [15,16]. Any ecological effect following antibiotic-BDI withdrawal will likely contribute to the spill-over effect [43]. Whether reversible antibiotic tolerance induced in *S. aureus* by exposure to colistin, a common constituent of antibiotic-BDI regimens, contributes to this emergence of Gram-positive isolates is unclear. This tolerance will be undetectable by standard susceptibility testing [44].

The high *S. aureus* VAP and *S. aureus* BSI incidences within the concurrent control groups of antibiotic-BDI studies noted here as a spill-over effect, which is as previously noted for several end points [8,17,18,19,23,43,45,46], would conflate the apparent prevention effect.

For antiseptic-BDI, the results of prevention studies may differ depending on the end point of interest, the patient risk category and LOS. For example, pneumonia prevention is noted among cardiac ICU patient populations, with short LOS, whereas possible harm occurs with medical ICU patient populations, in association with long LOS [13,14].

### Limitations

Several limitations should be considered. In meeting objective one, the indicative summary estimates derived here are comparable with those elsewhere in the literature (Table 1 and Table 4). Given the uncertain amount of spillover effect in these concurrent controlled decontamination studies, achieving unbiased patient level effect estimates will be elusive and here are designated indicative. In meeting the other objectives, group-level estimates were derived from broadly selected studies from the literature, not limited to those that were randomized, blinded or even controlled. Hence, the literature search has been opportunistic rather than systematic. By using existing systematic reviews as a starting point, the key interventions can be readily identified and classified.

In meeting objective two, there is considerable heterogeneity in the interventions, populations, modes of VAP diagnosis, study quality and study designs among studies published over several decades included in the analysis here with no ability to adjust for underlying patient risk. Hence, these effect estimates are also considered ‘indicative’ and primarily relate to the group level rather than the patient level of analysis. The definitions of VAP used in various studies vary and this further adds to the heterogeneity for endpoints related to VAP incidence.

In meeting objective three, the use of broad inclusion criteria was intentional as this objective required estimating the incidence of *S. aureus* infections in cohorts with various exposures to interventions, spillover and LOS. This requires a benchmark derived from non-interventional (observational) cohorts for comparison. Studies with a non-concurrent control (NCC) design are also of great interest as they will be free from any spill-over effect from any intervention concurrently under study in the ICU, in contrast to the case in studies where control and intervention groups are concurrent. NCC studies are usually excluded from Cochrane reviews which rate these studies with lower quality scores. However, the potential effects of spillover and rebound are not recognized in the scoring of study quality.

Of note, there were no obvious individual study result outliers driving the overall findings despite the heterogeneity among the results of studies included in accord with broad selection criteria.

Mean LOS and even median LOS are crude measures of group-level exposure of each group to the ICU context and exposure to the infection prevention interventions in the intervention groups. The analysis is ecological, and the estimates relate to the impacts of antiseptic-BDI and antibiotic-BDI on ICU patient cohorts. Of note, even cohorts with short mean LOS will contain patients with long LOS and vice versa. The associations for group-wide exposures may not equate to associations at the patient level of exposure.

This analysis here is unable to determine either the duration of *S. aureus* spillover as a driver nor the relationship between the timing of the *S. aureus* rebound and the cessation of the antibiotic- or antiseptic-BDI, which for many studies was unclear.

Many studies of decontamination interventions will have been underpowered to adequately assess key safety end points. Moreover, none were able to assess for whatever microbiome interactions that might underlie the bacteremia prevention effect [19].

The GSEM is a group-level modelling of the latent variable, *Staphylococcus aureus* colonization, within a postulated model of causation which can accommodate all studies and both infection end points. This latent variable and the coefficients derived in the GSEM are indicative only. They have no counterpart at the level of any one patient or study. They indicate the propensity for invasive infection arising, by whatever mechanism, from colonization as a latent construct rather than colonization measured by its presence and density.

Finally, there is the potential for publication and reporting bias as studies finding a significant effect size are more likely to be published. Whilst *S. aureus* infection incidences were rarely the primary end point, those that have found significant differences for *S. aureus* infection incidences may be more likely to report these findings as secondary end points than if the differences were neutral or negative. Whilst the possibility of competing risks in estimating *S. aureus* infections, such as early exclusion due to ICU discharge or mortality, would likely be similar for the studies regardless of study intervention, the development of VAP would be expected to prolong the ICU stay, and this is likely reflected in the higher LOS among studies of antibiotic-BDIs.

## 4. Conclusions

Antibiotic-BDIs appear to prevent *S. aureus* VAP and *S. aureus* BSI, although this effect varies with mean study LOS. The prevention effect of antibiotic-BDIs against *S. aureus* VAP attenuates in association with group mean LOS. The prevention effect of antibiotic-BDIs against *S. aureus* BSI inverts in studies with mean LOS > 14 days due to rebound within the intervention groups. This rebound would explain the paradoxical findings in the literature.

## 5. Materials and Methods

Being an analysis of published work, ethics committee review of this study was not required.

### 5.1. Study Selection and Decant of Groups

The literature search used here is as described previously [23]. Cochrane reviews and other systematic reviews were used as the primary source of studies, with additional studies being found by snowball sampling using the “Related articles” function within Google Scholar [9,10,11,12,13,24,25,26,27,28,29,30,31,32,33,34,35,36,37].

The inclusion criteria were cohorts of patients requiring prolonged (>24 h) ICU stay for which either or both incidence proportions of *S. aureus* VAP or *S. aureus* BSIs and either mean or median LOS were reported. Where possible, data were extracted for each identifiable sub-cohort representing different patient types or observation eras from the studies.

The studies were classified into four broad groups of either non-decontamination, antiseptic-BDI or antibiotic-BDI and studies (observational) without an intervention under study. The fourth category served as the benchmark category in the meta-regression of *S. aureus* VAP or *S. aureus* BSI incidence. These observational studies were screened to exclude any in which an infection prevention intervention was under study.

Non-decontamination interventions were studies of various approaches to the control of upper gastro-intestinal tract colonization through various stress ulcer prevention or feeding approaches and various approaches to control airway colonization through airway management.

Antiseptic-BDI included use of agents such as chlorhexidine, povidone–iodine and iseganan. All antiseptic exposures were included regardless of whether the application was to the oropharynx, by toothbrushing or by body-wash. Antibiotic-BDI is the use of topical antibiotic prophylaxis (TAP) to the oropharynx or stomach without regard to the specific antibiotic constituents or whether protocolized parenteral antibiotic prophylaxis (PPAP) was used in addition as part of the antibiotic-BDI regimen. Note that mupirocin appears as a component within arms of one antiseptic-BDI and two antibiotic-BDI studies. The use of antibiotic therapy outside of that dictated by study protocols has not been factored into this study.

The inclusion criteria were deliberately broad without regard to the frequency or duration of intervention under study or any criteria of study quality.

### 5.2. Outcomes of Interest

The independent variable in the regression models was the mean length of ICU stay (LOS). If this was not available, the median LOS or the mean or median duration of mechanical ventilation (MV) were used. The *S. aureus* VAP, *S. aureus* BSI and LOS data were all derived from the original publications. The *S. aureus* VAP, *S. aureus* BSI and LOS data required transformation. The first two, being count data, were logit transformed. The *S. aureus* VAP incidence proportion is the proportion with *S. aureus* VAP using the number receiving prolonged (>24 h) MV as the denominator. The *S. aureus* BSI incidence proportion is the proportion with *S. aureus* BSI using the number of patients with prolonged (>24 h) ICU stay as the denominator. The LOS data are positively skewed and were transformed for all analyses as follows: any LOS < 4 days was truncated to 4 days; the LOS was divided by 7 and then log transformed. With this transformation, all model intercepts equated to group mean LOS 7 days.

### 5.3. Summary Effect Size

Indicative summary prevention effect sizes, versus each of *S. aureus* VAP and *S. aureus* BSI, for each category were derived from all studies regardless of whether the intervention was randomly assigned and whether study blinding was achieved. The summary prevention effect sizes versus overall VAP and overall BSI were also derived from the same studies where available.

The study-specific and summary prevention effect sizes of each of the three broad categories of intervention toward the prevention of *S. aureus* VAP and *S. aureus* BSI incidence were generated by random effects using the meta-analysis command in Stata (Stata 18, College Station, TX, USA) [47].

### 5.4. Meta-Regression Model 1: Study Effect Size versus LOS

Models of the relationship between study-specific prevention effect sizes versus log transformed LOS of each of the three broad categories of intervention toward the prevention of *S. aureus* VAP and *S. aureus* BSI incidence were generated by meta-regression. The estimated effect sizes for studies with mean LOS of 7 and 20 days were derived post estimation using the margins command in Stata.

### 5.5. Meta-Regression Model 2: S. aureus VAP and S. aureus BSI Incidences versus LOS

Models of the relationship between logit transformed *S. aureus* VAP and *S. aureus* BSI incidence proportion versus log transformed LOS were generated using the meta-regress command in Stata. Scatter plots also were generated to facilitate a visual summary. In the scatter plots, the linear regression derived for each of the transformed *S. aureus* VAP and *S. aureus* BSI incidence versus the transformed LOS among observational cohorts were used as the respective benchmarks. The estimated *S. aureus* BSI and *S. aureus* VAP incidence for studies with a mean LOS of 7 and 20 days were derived post estimation using the margins command in Stata.

### 5.6. Generalized Structural Equation Modelling

Generalized structural equation modelling (GSEM) methods are an extension of SEM methods applied to count data. In the GSEM models, the VAP and BSI incidence proportion data serve as the measurement components, the group-level exposure parameters serve as the indicator variables and *S. aureus* colonization, being represented as a latent variable, links the indicator and measurement components in the model. In these models the antibiotic-BDI are factorized into TAP and PPAP components. The GSEM methodology used here is described in detail in previous publications [19,48].

Study identifiers were used in the models to enable the generation of robust variance covariance matrices of the coefficient estimate parameters of observations clustered by study. The ‘GSEM’ command in Stata (Stata 18, College Station, TX, USA) was used.

### 5.7. Availability of Data and Materials

All data generated or analyzed during this study are included in this published article and its Appendix A (see ESM).

## Figures and Tables

**Figure 1 antibiotics-13-00316-f001:**
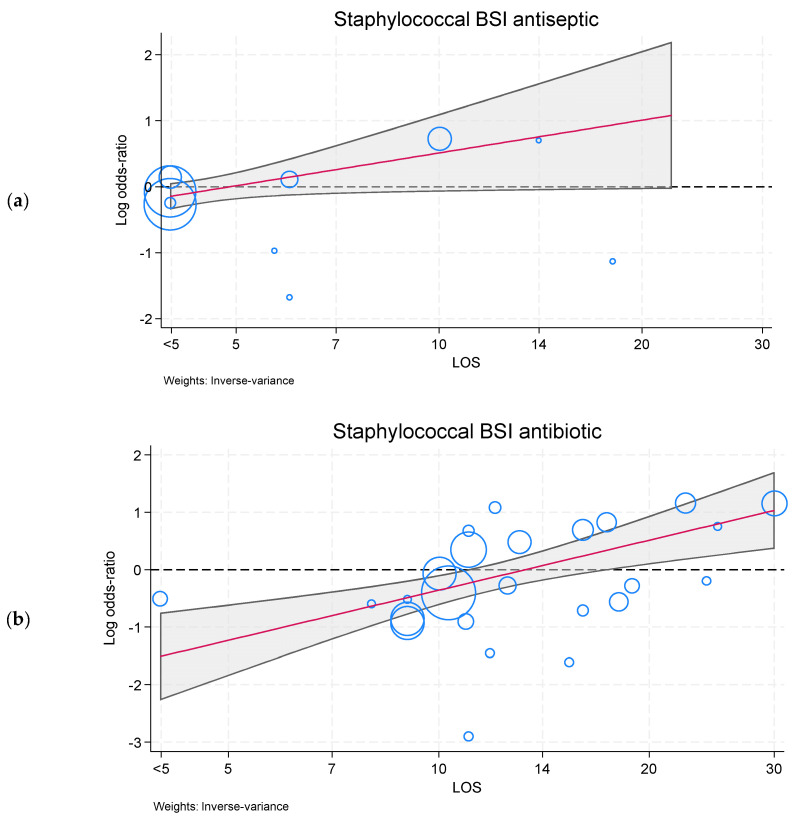
Meta-regression (red line with shaded 95% confidence intervals) of effect size (as log odds ratio, with individual studies appearing as blue circles with size proportional to the inverse variance) of antiseptic-BDI (**a**) and antibiotic-BDI (**b**) in preventing *S. aureus* BSI versus group mean LOS (logarithmic scale). Meta-regression plots for non-decontamination category of interventions in preventing *S. aureus* BSI (Appendix A) and all three intervention categories in preventing *S. aureus* VAP are presented in the online supplement as figures (Appendix A) and as a summary table (Appendix A).

**Figure 2 antibiotics-13-00316-f002:**
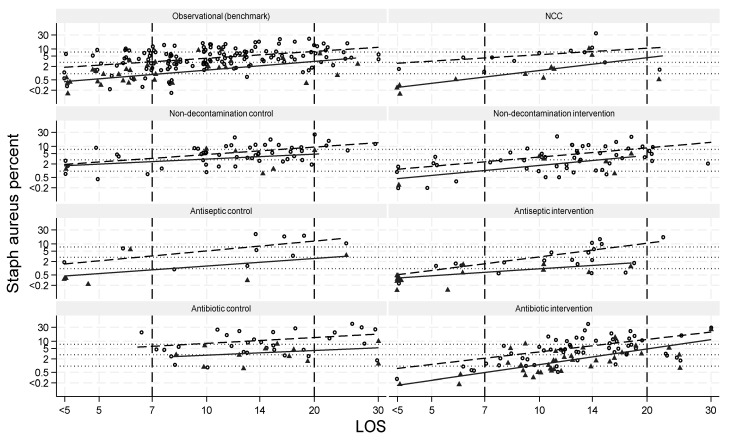
Meta-regression of *S. aureus* VAP (○, broken regression line) and BSI (▲, solid regression line) incidence among groups of ICU patients within studies of various prevention interventions versus group mean LOS (logarithmic scale). The dotted horizontal lines at 1 and 3 percent are the day 7 and day 20 intercepts for the *S. aureus* BSI (as derived in Appendix A) and the dotted horizontal lines at 3 and 7.6 percent are the day 7 and day 20 intercepts for the *S. aureus* VAP (as derived in Appendix A), each as derived in the meta-regression model for observational groups. The regression model coefficients are presented in Appendix A. The NCC (non-concurrent control) category includes control groups that, being non-concurrent to antibiotic-BDI or antiseptic-BDI groups, were not exposed to spillover effects from these interventions. Note that the y-axis is a logit scale and the x-axis is the group mean LOS transformed to a logarithmic scale after dividing by 7 such that the model intercept values correspond to group mean day 7 estimates as derived in the meta-regression models.

**Figure 3 antibiotics-13-00316-f003:**
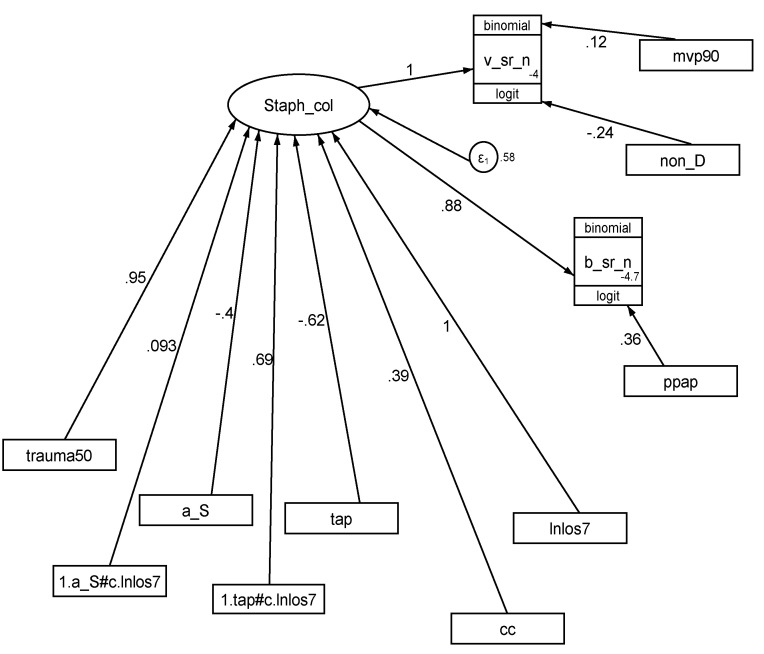
GSEM of *Staphylococcus* colonization. *Staphylococcus* col (oval) is a latent variable representing *Staphylococcus* colonization. The variables in rectangles are binary predictor variables representing the group-level exposure to the following: trauma ICU setting (trauma50), mean or median length of ICU stay transformed by dividing by 7 and logarithmic transformation (lnLOS7), exposure to a topical antiseptic-BDI (a_S), exposure to an antibiotic-BDI (tap being topical antibiotic prophylaxis as a component of antibiotic-BDI), exposure to a non-decontamination-based prevention method (non-D), exposure to protocolized parenteral antibiotic prophylaxis (ppap) and more than 90% of the cohort receiving mechanical ventilation (mvp90). In this model, the effect of spillover equates to the effect of concurrency of a control group with an antiseptic or antibiotic-based BDI intervention group (CC), and the effect of rebound equates to the interaction term between LOS and exposure to antibiotic- (1.tap#c.lnlos7) or antiseptic-BDI (1.a_S#c.lnlos7). Note that the model factorizes exposures from compound regimens (e.g., SDD and SOD, which combine topical antibiotic prophylaxis [TAP] with or without PPAP as an antibiotic-BDI) into singleton TAP and PPAP exposures. The circle represents the model error term. The three-part boxes represent the binomial proportion data for *Staphylococcus* VAP (v_sr_n) and BSI (b_sr_n) counts with the number of patients as the denominator which is logit transformed using the logit link function in the generalized model.

**Table 1 antibiotics-13-00316-t001:** Characteristics of studies ^a^.

	Observational Studies	Infection Prevention Studies		
	(No Intervention)	Non-Decontamination	Antiseptic-BDI	Antibiotic-BDI
Study characteristics				
Listing	Appendix A	Appendix A	Appendix A	Appendix A
Number of studies	145	53	26	60
Origin from systematic review ^b^	46	20	14	27
North American ICU’s ^c^	33	12	8	2
MV for >48 h for <90% ^d^	24	0	9	4
Control group PPAP ^e^	2	0	0	2
Group mean age; (mean, 95% CI)	56.1 54.4–57.5	55.4 52.3–58.4	48.5 42.0–54.5	50.1 46.2–54.0
Study publication year (range)	1986–2023	1987–2022	2000–2015	1985–2022
Group characteristics				
Types of groups				
Observational	165			
NCC	NA ^f^	0	16	22
CC	NA	53	11	30
Intervention	NA	54	75	152
Numbers of patients per group; (median; IQR) ^g^	300 135–936	81 61–155	72 31–347	54 34–84
Group mean LOS; (mean, 95% CI)	10.8 9.9–11.6	12.9 11.3–14.4	11.4 8.3–14.5	15.9 14.0–17.9
Overall prevention effect size; ^h^				
Total VAP (odds ratio; 95% CI; number of studies)	NA	0.69; 0.59–0.82 (54)	0.65; 0.48–0.89 (17)	0.34; 0.27–0.42 (46)
Total bacteremia (odds ratio; 95% CI; number of studies)	NA	0.96; 0.64–1.44 (8)	0.73; 0.61–0.87 (13)	0.62; 0.52–0.74 (39)
*S. aureus* prevention effect size;				
*S. aureus* VAP (odds ratio; 95% CI; number of studies)	NA	0.76; 0.65–0.90 (51) (see Appendix A)	0.44; 0.30–0.64 (13) (see Appendix A)	0.57; 0.46–0.7 (40) (see Appendix A)
*S. aureus* bacteremia (odds ratio; 95% CI; number of studies)	NA	0.66; 0.31–1.4 (5) (see Appendix A)	0.98; 0.74–1.3 (10) (see Appendix A)	0.92; 0.65–1.29 (25) (see Appendix A)

^a^. Note that several studies had more than one control and/or intervention group. Hence, the number of groups does not equal the number of studies; ^b^. Studies that were sourced from 16 systematic reviews (references in web-only supplementary); ^c^. Study originating from an ICU in Canada of the United States of America; ^d^. Studies for which less than 90% of patients were reported to receive >48 h of MV; ^e^. Use of protocolized parenteral antibiotic prophylaxis (PPAP) for control group patients; ^f^. NA—not applicable; ^g^. Data are median and inter-quartile range (IQR); ^h^. Effect sizes for total VAP and BSI prevention includes data from all studies for which this effect size could be calculated whether or not *S. aureus* VAP or BSI data were available.

**Table 2 antibiotics-13-00316-t002:** Predicted *S. aureus* infection incidence per 100 patients from meta-regression models versus LOS ^a^.

	Day 7		Day 20	
	Prediction	95% CI	Prediction	95% CI
*S. aureus* VAP				
Observational (Benchmark)	3.0	2.5–3.6	7.6	5.9–9.1
Non-concurrent control	3.6	2.1–6.3	8.3	3.6–18.2
Non-decontamination control	3.2	2.2–4.7	9.1	6.3–11.9
Non-decontamination intervention	2.4	1.6–3.6	7.6	5.1–10.9
Antiseptic-BDI control	2.9	1.2–6.3	13.0	6.2–25.0
Antiseptic-BDI intervention	1.3	0.7–2.4	9.1	4.8–16.8
Antibiotic-BDI control	6.9	3.4–13.0	11.9	8.5–16.8
Antibiotic-BDI intervention	1.6	1.1–2.7	10.0	7.1–14.2
*S. aureus* BSI				
Observational (benchmark)	1.1	0.9–1.5	3.2	1.9–4.7
Non concurrent control	0.7	0.5–1.2	3.9	1.6–9.9
Non-decontamination control	2.0	0.9–4.7	2.4	0.9–5.7
Non-decontamination intervention	1.1	0.3–3.6	2.4	1.0–6.3
Antiseptic-BDI control	1.1	0.5–2.4	3.9	0.6–19.8
Antiseptic-BDI intervention	0.7	0.4–1.2	2.7	0.5–11.9
Antibiotic-BDI Control	1.6	0.5–5.7	3.6	2.0–6.3
Antibiotic-BDI intervention	0.6	0.4–1.0	6.3	4.0–9.1

^a^. Predicted *S. aureus* VAP and *S. aureus* BSI incidences derived from meta regression models as presented in Figure 2 [as the broken and solid regression lines, respectively]. Appendix A contains the slope and intercept coefficients of these models.

**Table 3 antibiotics-13-00316-t003:** GSEM model ^a,b^.

	Model 2 (Appendix A)	Model 1 (Figure 3)	
Factor (Label Abbreviations as in Figure 3 and Appendix A)	Coefficient	Coefficient	95% CI
b_Sr_n ^c^			
Staphylococcal colonization	0.88 ***	0.88 ***	0.75 to 1.01
ppap ^d^	0.33 *	0.36	−0.15 to 0.86
_ Constant	−4.75 ***	−4.72 ***	−4.9 to −4.54
v_Sr_n ^c^			
Staphylococcal colonization	1	1	(constrained)
mvp90 ^d^	0.18	0.12	−0.23 to 0.47
non_D ^d^	−0.25	−0.24	−0.53 to 0.05
_ Constant	−4.1 ***	−4.0 ***	−4.64 to −3.3
Staphylococcal colonization ^c^			
CC (Concurrency to TAP use) ^e^	0.36 *	0.39 *	0.05 to 0.73
TAP ^d^	−0.25 *	−0.62 **	−1.07 to −0.18
TAP * LnLOS7 (interaction) ^f^		0.69 *	0.09 to 1.29
Antiseptic	−0.39 *	−0.40 **	−0.69 to −0.11
Antiseptic * LnLOS7 (interaction) ^f^		0.09	−0.41 to 0.60
LnLOS7 ^g^	1.07 ***	1.00 ***	0.15 to 0.73
Trauma50 ^h^	0.97 ***	0.95 ***	0.72 to 1.18
Variance (e. Staph col)	0.59 ***	0.58 ***	0.46 to 0.73
Groups (n)	442	442	
Clusters (n)	277	277	
Factors	13	15	
Akaike Information Criteria (AIC)	3240	3238	

^a^. Legend: * *p* < 0.05; ** *p* < 0.01; *** *p* < 0.001. ^b^. Shown in this table are the models corresponding to Figure 3 and Appendix A. ^c^. v_sr_n is the count of *Staphylococcal* VAP; b_sr_n is the count of *Staphylococcal* bacteremia; and Staph col is Staphylococcal colonization (*Staphylococcal* col; which is a latent variable). ^d^. PPAP is the group-wide use of protocolized parenteral antibiotic prophylaxis; non-D is a non-decontamination intervention; MVP90 is use of mechanical ventilation by more than 90% of the group for >24 h; TAP is topical antibiotic prophylaxis; ^e^. The effect of concurrency to TAP use equates to spillover. ^f^. Rebound equates to the interaction term between LOS and exposure to antibiotic- (1.tap#c.lnlos7) or antiseptic-BDI (1.a_S#c.lnlos7). ^g^. LnLOS7 is Length of stay transformed by dividing by 7 and logarithmic transformation. ^h^. Trauma50 is an indicator variable for those groups with a majority of patients admitted for trauma.

**Table 4 antibiotics-13-00316-t004:** Comparison with previous prevention effect size estimates.

	Antiseptic-BDI				Antibiotic-BDI			
	Coefficient	95% CI	n ^a^	Ref.	Coefficient	95% CI;	n ^a^	Ref.
VAP or RTI (Overall)								
RTI	RR: 0.76	0.62 to 0.91	18	[27,28]	OR: 0.28	0.2 to 0.38	16	[24]
VAP	OR: 0.68	0.53 to 0.87	21	[9]	RR: 0.43	0.35 to 0.53	17	[25]
NP/RTI	OR: 0.66	0.51 to 0.85	22	[9]	RR: 0.44	0.36 to 0.54	22	[26]
NP/RTI	RR: 0.73	0.58 to 0.92	16	[13]				
VAP	OR: 0.65	0.48 to 0.89	17	This study	OR: 0.34	0.27 to 0.42	46	This study
BSI (Overall)								
BSI	OR: 0.74	0.37 to 1.5	5	[12]	RR: 0.68	0.58 to 0.80	21	[26]
BSI					OR: 0.73	0.59 to 0.90	31	[11]
BSI	OR: 0.73	0.61 to 0.87	13	This study	OR: 0.62	0.52 to 0.74	39	This study
Gram-positive VAP or RTI								
Gram-positive RTI	OR: 0.41	0.19 to 0.85	9	[9]	OR: 0.52	0.34 to 0.78	14	[10]
MSSA RTI	OR: 0.48	0.2 to 1.16	21	[9]				
*S. aureus*	OR: 0.44	0.3 to 0.64	13	Appendix A; this study	OR: 0.57	0.46 to 0.7	40	Appendix A; this study
Gram-positive BSI								
Gram-positive BSI	OR: 0.72	0.23 to 2.22	3	[12]	OR: 1.03	0.75 to 1.41	19	[10]
Gram-positive BSI					OR: 1.06	0.77 to 1.47	16	[11]
*S. aureus*	OR: 0.98	0.74 to 1.3	10	Appendix A; this study	OR: 0.92	0.65 to 1.29	25	Appendix A; this study

Abbreviations: OR, odds ratio; RR, risk ratio; 95% CI, 95% confidence interval; RTI = respiratory tract infection; NP = nosocomial pneumonia; VAP = ventilator-associated pneumonia; BSI = blood stream infection; MSSA = methicillin susceptible *S. aureus*. ^a^. n = number of studies

## Data Availability

The datasets analyzed during the current study are provided in the Appendix A.

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
