# Peer review of "Rebound Inverts the Staphylococcus aureus Bacteremia Prevention Effect of Antibiotic Based Decontamination Interventions in ICU Cohorts with Prolonged Length of Stay"

_antibiotics, 2024, doi:10.3390/antibiotics13040316_

Round 1

Reviewer 1 Report

Comments and Suggestions for Authors

The content of the study is very interesting and has an high impact on medical practice in hospitals, in the prevention of hospital acquired infections and therapy. I suggest for the authors to continue their study taking in the account the importance of detection and decontamination also for other bacteria.

Author Response

Response to reviewer 1.

I thank reviewer 1 for these kind comments.  

As noted by this reviewer, I have performed studies taking in the account the importance of detection and decontamination for other bacteria. These are references 17-19, 43 & 46.

Reviewer 2 Report

Comments and Suggestions for Authors

This study seeks to determine the effects on S aureus, VAP and blood stream infection through analysing published work. The definition of VAP is difficult and may vary between studies, but this is not discussed. Decontamination interventions (BDI) mention antibiotics but are usually based on topical disinfection with chlorhexidine and mupirocin aimed given generally or selectively against MRSA/S aureus. Was mupirocin under the antibiotic or antiseptic intervention group? There is no explanation of the range of BDI types included other than they are either antiseptic, or antibiotic based. As they are usually given together how were effects separated? Why were some groups given only iv antibiotic prophylaxis as a protocol risking resistance emergence – were these ICUs using selective decontamination?

Studies that were mentioned in Cochrane reviews should be examined first hand rather than taking figures from Cochrane. More explanation is needed to explain the length of stay effects other than showing more infections occur over time. Control groups that were non-concurrent are a potential problem because infection control activities continue over time independent of research work. In Table 2 what is ‘precited infection incidence’? What are the units – possibly /100 patients? Why would intervention groups have higher incidence of infection at 20 days than controls with no intervention? Does previous decolonization predispose to later colonization through emergence of resistance?

The authors assume a high level of knowledge of modelling methodology and provide little explanation for example Akaike Information Criteria. Background explanation would aid understanding. Numerous abbreviations are used which do not help. The etiology of VAP and BSI overlap but are not the same. Catheter infections may be caused by introduction of S aureus directly from exogenous sources.  By the time of the observed inversion were interventions no longer being used? The spillover effect is mentioned as a possible explanation but how long did this continue during each study and how important was it as usually studies have safeguards in the protocol. Non-concurrent controls are subject to other sources of potential bias over time.

Author Response

Response to reviewer 2.

I thank reviewer 2 for these comments which are preceptive and very helpful. 

Discussion that the definition of VAP is difficult and may vary between studies is noted in the second paragraph of ‘4.1 Limitations’ [now ‘3.1 Limitations’]. I have expanded this point.

Mupirocin has been used both within studies of topical antibiotics and studies of topical anti-septics. There are relatively few study arms containing mupirocin intervention among 136 studies and to simplify, these have been classified as per the systematic review from which the study was derived from. A comment has been added within the materials and methods.

There was no attempt to separately estimate the effects of specific antiseptic, or antibiotic components. Rather the objective is to estimate the effect of rebound regardless of the specific broadly classified Antiseptic or antibiotic BDI. I have expanded on this point in the introduction and the discussion [as the fourth objective].

The reviewer notes that some groups given only iv antibiotic prophylaxis as per protocol within ICUs using selective decontamination. This is correct and underlines the need to separately account for this effect of this component which is only possible within a GSEM modeling approach. A comment has been added within the discussion.

The studies that were mentioned in Cochrane reviews have been examined first hand - A comment has been added within the materials and methods.

I am unsure how to explain the length of stay effects other than showing more infections occur over time nor the effect of whether infection control activities continue over time independent of research work. Comments within ‘4.1 Limitations’ [now ‘3.1 Limitations’] respond to these comments.

In Table 2, the ‘predicted [typo corrected] infection incidence’ is the dotted and solid regression lines of Fig 2 as indicated in the footnote [now clarified]. The units are per 100 patients [clarified in Table heading]. That intervention groups have higher incidence of infection at 20 days than controls with no intervention is a consequence of rebound of infection and spillover between groups. This is a major paradox which is not inconsistent with the effect size appearing to indicate infection prevention. Note, this is rebound rather than colonization through emergence of resistance. A comment has been added within the discussion.

The explanation for the modelling methodology, for example Akaike Information Criteria, is provided in the companion references. A comment has been added in the methods. The abbreviations are required [as is Fig 3].

Whilst the etiology of VAP and BSI overlap, of note, it is the origin from colonization that is of interest here [as noted in the introduction].

The timing of the cessation of interventions in relation to the inversion is unclear. Also, how long spillover might continue is unclear. Comment has been added within the discussion.

Whilst non-concurrent controls are subject to other sources of potential bias over time, the incidences of S aureus VAP and BSI infections are similar to that in the observational groups.

Reviewer 3 Report

Comments and Suggestions for Authors

Hello:

Thanks for giving me an opportunity to review. It is a good study however certain things aren't clear to me as reviewer. 

- Did you adjust for outliers in the effect sizes?

- In the supplementary tables, can you add the effect sizes for each study?

- I understand you wished to focus on meta regression but can forest plots and funnel plots (to account for publication bias) be provided?

- The tables should be formatted well. 

- I am a bit confused. If you are calculating incidence (Which I assume follows a negative binomial) and you are using those receiving MV as offset why are you reporting OR in Figure 1? Shouldn't it be RR. 

- Rather than providing a table in the discussion section, please write the major findings. 

Author Response

Response to reviewer 3.

I thank reviewer 3 for these comments. 

To answer these questions.

  • There was no adjustment for outliers in the effect sizes. This could be done as a sensitivity test if there was one influential outlier study in the overall results but I did not identify one.
  • Whilst the focus is on meta-regression, forest plots have also been provided [Fig s1 – s3]. I have not provided funnel plots as this would be a distraction to the meta-regression. Note that these forest plots are formatted as caterpillar plots to enable the detection of any outlier results [ there were none]. There is a comment relating to publication bias as a limitation [last paragraph of 3.1 beginning “Finally….”].
  • I have reformatted the tables to make them clearer for the reader.
  • Please note that Figure 1 is a meta-regression of effect size expressed as odds ratios [as per standard practice] whereas Figure 2 presents the incidence in which there is no need for offset or a negative binomial.
  • I have provided the table in the discussion to provide a succinct summary of previous study results to aid reader comprehension and to address objective 4. As suggested by this reviewer, I have also provided a commentary on these findings.

Round 2

Reviewer 2 Report

Comments and Suggestions for Authors

Thank you for making revisions. However I am still unclear on the authors understanding of the concept of rebound. Do you mean that colonizing S aureus is suppressed whilst topical suppression or prophylactic antibiotics are used and then regrow once stopped? This is to be expected and might result in further infection. Is the duration of prophylactic measures available and can it be related to the findings? Prophylactic measures particularly topical measures do not eradicate colonizing staphylococci completely. Patients staying longer in hospital are subject to further nosocomial infection.

Author Response

  • Yes, colonizing S aureus is suppressed whilst topical suppression or prophylactic antibiotics are used and then regrow once stopped. Whilst this is to be expected and it might result in further infection, this is occurring in a microbiome that has been altered by the exposure to the ABD as well as the altered microbiome of the ICU. I have clarified at line 47 & 341-342.
  • Unfortunately, the duration of prophylactic measures is not available and hence cannot be related to the findings. This is a stated limitation at lines 337-339.
  • Yes, prophylactic measures particularly topical measures do not eradicate colonizing staphylococci completely. This is part of the rebound process. I have clarified at line 56 and also at lines 338-342.
  • Yes, patients staying longer in hospital are subject to further nosocomial infection. This is the reason for the meta-regression of the incidences versus group mean LOS [as stated at line 31-32; line 61-62; in table s6 & all of section 2.4 and in in the methods]. This underlies the results described in sections 2.3 & 2.4 but is most apparent for S aureus BSI [which I now highlight at line 265-269].

Reviewer 3 Report

Comments and Suggestions for Authors

Hello:

- Thanks for your quick reply. Thanks for indicating the forest plots clearly. They were hidden in the supplementary tables.

- As you have modified the tables they are definitely easier to read.

- Can you add a line or two about the outlier analysis that you have performed. 

Author Response

To answer these questions.

  • I have added a note that there were no outlier results [line 329-331].

Round 3

Reviewer 2 Report

Comments and Suggestions for Authors

No further issues